# Recent Advances in Cellulose-Based Hydrogels Prepared by Ionic Liquid-Based Processes

**DOI:** 10.3390/gels9070546

**Published:** 2023-07-05

**Authors:** Siriporn Taokaew

**Affiliations:** Department of Materials Science and Bioengineering, School of Engineering, Nagaoka University of Technology, Nagaoka 940-2188, Niigata, Japan; t.siriporn@mst.nagaokaut.ac.jp

**Keywords:** cellulose, ionic liquid, hydrogel, regeneration, conductive hydrogel, circular economy

## Abstract

This review summarizes the recent advances in preparing cellulose hydrogels via ionic liquid-based processes and the applications of regenerated cellulose hydrogels/iongels in electrochemical materials, separation membranes, and 3D printing bioinks. Cellulose is the most abundant natural polymer, which has attracted great attention due to the demand for eco-friendly and sustainable materials. The sustainability of cellulose products also depends on the selection of the dissolution solvent. The current state of knowledge in cellulose preparation, performed by directly dissolving in ionic liquids and then regenerating in antisolvents, as described in this review, provides innovative ideas from the new findings presented in recent research papers and with the perspective of the current challenges.

## 1. Introduction

The development of products with durability, repairability, and reusability is a sustainable way to mitigate environmental impacts and move toward a circular economy. The impact on the environment has manifested from the continuous use of petroleum-based resources that must be substituted with alternative renewable and eco-friendly resources [1,2]. Biomass-based raw materials are a promising replacement of non-renewable, petroleum-based materials for advancing the efforts toward achieving a sustainable society. In order to achieve the desired levels of sustainable growth and to mitigate environmental pollution, biomass has been applied to reduce wastes and develop value-added products [3]. Expanding the use of such biomass-based materials does not only address the societal need for low-cost materials, but also reduces carbon footprints [4,5]. As a consequence of the abundance of renewable biomass materials in nature, its rational use is of great significance for the sustainable development of the chemical industry [6].

Cellulose is a biomass source with the potential to replace fossil fuels through its use in the production of value-added chemicals such as 5-hydroxymethylfurfural and levulinic acid [7]. It is a natural polymer available from diverse sources such as plants and from the extracellular production of several microbial genera, particularly *Gluconacetobacter xylinus* [8,9]. Cellulose can be obtained from wastes such as paper [10], textile [11,12,13], oil palm frond [14], and crop straw [15,16]. Due to its availability, biodegradability, non-toxicity, hydrophilic nature, inexpensive cost, and multifunctionality, cellulose is attractive and shows promise for utilization in sustainable material engineering. The processibility of cellulose allows it to be formed into various types of materials, for example, into hydrogels [17] and/or applied in its dry form as aerogels [18]. Cellulose hydrogels (jelly-like solids) are obtained by the regeneration or crosslinking of cellulose in solutions into a 3D shape, which are different from the non-cross-linked gel-like suspension of nano- or micro-fibrillated cellulose [19,20,21]. Due to the presence of hydroxyl groups in cellulose, the cross-linking in the hydrogel is formed through hydrogen bonding, which is reconstructed during the cellulose regeneration and solvent removal process [22]. Numerous hydroxyl groups in cellulose and other hydrophilic groups, e.g., carboxyl and amino groups, present in the cellulose derivatives, enable the entrapment of large volumes of water [23,24]. Depending on their applications in biotechnology, cosmetology, and ecology, a variety of sizes, from macro to nanogels, such as membranes, films, fibers, microspheres, or nanoparticles, can be fabricated by various interactions [25,26]. Those interactions are hydrogen bonding, electrostatic interactions, van der Waals forces, and physical entanglements of the cellulose molecules and/or their blends [25,26]. One of the main limitations of processing native cellulose is that it is difficult to dissolve in conventional organic or inorganic solvents due to the abundance of hydrogen bonds. The cellulose derivatives, e.g., carboxymethylcellulose, are much easier to dissolve, but multiple production steps are required to produce such derivatives [27]. The native cellulose is a semi-crystalline polymer comprising a high order of crystalline regions and a lower order of amorphous regions [28]. The rich hydrogen bonds between and within cellulose polymer chains are beneficial for the mechanical properties of cellulose crystals which greatly enhances mechanical strength of highly crystalline cellulose materials [29]. However, the resultant strong and complex 3D inter- and intramolecular hydrogen bonding network contributes to the chain rigidity and insolubility of cellulose, which must be disrupted for dissolution [30,31]. The bonding disruption can be achieved by the derivatization of hydroxyl groups of cellulose or comparative formation of a stronger hydrogen bond with other strong hydrogen-bonding solvents [30,31]. The solvents that have been used in the sol–gel process of cellulose are: N-dimethylacetamide (DMAc)/LiCl [32]; tetrabutylammonium fluoride/dimethyl sulfoxide (TBAF/DMSO) [33]; alkali/urea (NaOH/urea/H_2_O, LiOH/thiourea/H_2_O, etc.) [34,35]; N-methyl morpholine oxide (NMMO) [36,37]; NMMO/DMSO [38]; ZnCl_2_ aqueous solutions [39,40,41,42]; ZnCl_2_ molten salt hydrates [43]; molten hydrates of LiClO_4_ NaSCN/KSCN/LiSCN and LiCl/ZnCl_2_ [44]; molten salt hydrates of ZnBr_2_ and FeCl_3_ [45]; LiBr trihydrate molten salt [46]; and ethylenediamine solutions of NaSCN, KSCN, or NaI [44,47]. However, harsh dissolution conditions, such as high temperature, strong corrosiveness from the alkali metal hydroxide, and long processing time, have been reported [48]. Direct non-derivatizing dissolution of cellulose by the NMMO-based Lyocell process is commercialized. Although it is less-time consuming, the thermal stability and side reaction of NMMO, which decreases the performance of hydrogel fibers, remains a concern [49]. The ZnCl_2_ aqueous solutions and inorganic molten salt hydrates have been of interest as cellulose dissolution solvents in recent years due to their ease of preparation and low cost [41,42,50]. However, the solubility of cellulose in such inorganic salts is low and requires a high temperature or long dissolution time, causing energy consumption and hydrolysis [50,51].

Ionic liquids (ILs) are considered to be desirable eco-friendly solvents and have been used to replace the organic solvents for cellulose dissolution under milder conditions. A regular crystalline structure is difficult to form because of the asymmetry of ILs and the reduced electrostatic forces between the cations and anions [52]. Therefore, they can be liquid at low or room temperature. The ILs can form comparatively stronger hydrogen bonding with the native molecular hydrogen bond network within cellulose structures, causing hydrogen bonding disruption and dissolution. Due to their main advantages besides their high cellulose dissolution ability, ILs have low vapor pressure and are efficiently recycled in order to avoid their propagation in ecosystems. Cellulosic solutions in ILs can also be converted into ionogels or hydrogels by simple operations such as regeneration in cellulose antisolvents [53]. Based on the simplicity and eco-friendliness of IL-based methods, this review summarized the findings of the research conducted in the past decade on fabricating cellulose hydrogels by using processes under pure IL, IL/cosolvent, and other new IL systems. Regarding the importance of continuous application for the green solvent, IL, in the direct dissolution process, the cellulose hydrogels and related products are truly sustainable according to the circular economy because their processing is also sustainable. 

## 2. Ionic Liquids Used in Dissolution of Cellulose 

### 2.1. Pure Ionic Liquid Systems

Anions and cations of ILs strongly affect cellulose solubility. The ILs containing cations with an unsaturated heterocyclic ring have the ability to dissolve cellulose because of their π electron delocalization of the ring. Such cations are active to interact with cellulose by providing more space for anions to form hydrogen bonds with cellulose. In contrast, the larger volume of ILs containing cations with the saturated heterocyclic ring slows down cations and anions transfer, which is less favorable for cellulose dissolution [54]. The options for cellulose dissolution have been ILs with aromatic imidazolium or pyridinium cations [55,56]. The imidazolium salts, such as 1-allyl-3-methyl imidazolium chloride (AmimCl), 1-buthyl-3-methyl imidazolium chloride (BmimCl), and 1-ethyl-3-methylimidazolium acetate (EmimAc), have received attention [57,58]. A series of substituted imidazolium cations, paired with lactate or glycolate anions, was developed by Meenatchi et al. [59]. For 5% *w*/*w* of cellulose solution, 1-ethyl imidazolium lactate showed the highest dissolution power within 20 min at 80 °C because of its higher number of acidic protons (C-2, C-4, C-5) which, in turn, favor the formation of more hydrogen bonds with oxygen atoms of cellulose. The replacement of the acidic C-2 proton by a methyl group (2-methylimidazolium) reduced the dissolution power, which suggested the substitution decreased the hydrogen bond interactions [59]. 

The anions also play a vital role in the breakage of the native cellulose hydrogen bond network because of the strong electronegativity, allowing anions to form strong hydrogen bonds with the hydroxide groups of cellulose. After being combined with various cations possessing alkyl functional groups, Cl^−^, acetate (Ac^−^), dimethyl phosphate (DMP^−^), and diethyl phosphate (DEP^−^) anions (acidic protons) strongly exhibit the ability to dissolve cellulose [60]. In contrast, weak hydrogen-bonding anions, e.g., BF_4_^−^, PF_6_^−^, and Tf_2_N^−^, are not suitable for cellulose dissolution [61,62]. The Cl^−^-based ILs are widely adopted as a small-sized hydrophilic hydrogen bond acceptor, for instance BmimCl is frequently used due to its thermoplasticity and thermal processability [63,64,65,66]. The EmimAc provides better solubility for cellulose due to its lower viscosity (~140 mPas at 25 °C and 10 mPa s at 80 °C) and to the fact that it forms easily processable and stable cellulose solutions; it is also less toxic (LD50 > 2000 mg/kg) and less corrosive than BmimCl [67,68,69,70]. Therefore, it has been a choice for creating regenerated cellulose hydrogels, especially in the form of fibers, because it requires less energy during the shaping process [71]. The AmimCl is also able to dissolve cellulose at a higher concentration than BmimCl [55,72], owing to its high Cl^−^ concentration, electrochemical stability, and compatibility of hydrogen bond acceptors providing more interaction sites for hydroxyl groups of cellulose [73,74]. AmimCl is an effective IL for cellulose dissolution and hydrogel fabrication. The regenerated cellulose produced from AmimCl has the highest crystallinity, tensile strength, and transparency compared with those from BmimCl and EmimAc [72]. Even though AmimCl shows greater cellulose dissolution capability than EmimDEP [75], the content of depolymerized cellulose increases more with time and temperature in AmimCl than in EmimDEP [75]. Xu et al. [76] studied Amim(MeO)PHO_2_ IL for cellulose dissolution and decomposition. To increase the ionicity of this IL, Na_2_PHO_3_ inorganic salt, having an anion structure similar to that of Amim(MeO)PHO_2_, was added to form a composite solvent system (Na_2_PHO_3_/Amim(MeO)PHO_2_) and facilitate cellulose dissolution. The Na_2_PHO_3_ addition reduced H^+^ concentration of the solvent system, which inhibited the decomposition of cellulose. The lower viscosity of Na_2_PHO_3_/Amim(MeO)PHO_2_ solvent caused 22% *w*/*w* cellulose to dissolve at 80 °C over 48 h. With the increase in Na_2_PHO_3_ content, the degree of polymerization and thermal stability of the regenerated cellulose increased, but the crystallinity decreased relative to that of the original microcrystalline cellulose (MCC), because the long molecular chains exhibited difficulty in forming new hydrogen bonds during regeneration [76].

One of the advantages of imidazolium-based ILs is their capability to dissolve cellulose even at room temperature. However, some drawbacks are the high viscosity and difficulty in cellulose dispersion due to the strong association between the anions and cations [77]. When ILs are used in the presence of lignin in the pulp or impurities, side reactions such as acetylation may occur at high temperatures above 100 °C [78]. Therefore, the researchers have studied new ILs for efficient cellulose dissolution. Li et al. [79] developed a series of superbase-derived ILs such as: 1,8-diazabicyclo [5.4.0] undec-7-ene (DBU) including 1,8-diazabicyclo [5.4.0] undec-7-enium (DBUH) carboxylate, e.g., OAc−; and 1,5-diazabicyclo [4.3.0] non-5-ene (DBN) including 1,5-diazabicyclo [4.3.0] non-5-enium (DBNH) carboxylate with lower viscosity and high cellulose solubility. The DBUH ethoxyacetate has the higher cellulose dissolution capacity (14.8% *w*/*w* wood pulp at 80 °C) [80], but lower corrosion to steels than EmimAc and AmimCl [81]. On the other hand, some organic N-oxide-based solutions can dissolve cellulose efficiently, but the drawbacks are the instability of the N-oxides and the high dissolution temperature leading to the partial degradation of cellulose. Lui et al. [82] prepared DBN coupled with two N-oxides, i.e., pyridine N-oxide (PyO) and 2-picoline-N-oxide (PiO). The solvents dissolved cellulose efficiently at temperatures lower than 80 °C. The solvent DBN/PyO, at molar ratio of DBN and PyO of four, showed the cellulose solubility of 10.9 and 14.1% *w*/*w* at 50 and 70 °C, respectively. Galamba et al. [83] studied benzethonium- and didecyldimethylammonium- based ILs combined with short alkyl carboxylate anions to dissolve MCC at a concentration of 4% *w*/*w*. The polymeric cellulose hydrogel structures varied depending on the type of ILs and the ratios between cellulose and IL. However, after regeneration, some ILs remained in the hydrogel structures, which was advantageous in terms of the antibacterial/antimicrobial response of the hydrogels. Some new IL systems for cellulose dissolution, as reported in recent studies, are shown in Figure 1.

### 2.2. Ionic Liquid–Cosolvent Systems

One of the problems with using ILs to process cellulose is the high viscosity of the cellulose/IL solutions. The use of IL/cosolvent mixtures is an option that can modify the properties of the IL solvent by improving the rheological properties of the cellulose mixture, to reduce the viscosity of the solvent system and to accelerate the mass transfer process. The solvation of cation and anion by the cosolvents enhances the production of more free anions from anion–cation pairs, which readily interact with cellulose [84,85]. The frequently used cosolvents are polar aprotic organic solvents like DMSO, dimethylformamide (DMF), dimethylacetamide (DMAc), and dimethylimidazolinone (DMI). Zhou et al. [86] studied the cellulose solution state and formation mechanism by using these four cosolvents with EmimAc, BmimAc, AmimCl, and BmimCl. Acetate-based ILs and a high cosolvent content facilitated the formation of a molecularly dispersed state. In contrast, a state of coexistence of single molecular chains and undissolved cellulose microdomains gradually converted into a molecularly dispersed state when using chloride based ILs or a low content of cosolvents. The cosolvents DMF and DMAc promoted the molecular dispersion of cellulose better than DMSO and DMI. In IL/cosolvent systems with a high hydrogen-bonding basicity (>0.92) and a large amount of small ion cluster structures, cellulose favors the formation of a molecularly dispersed state. Cellulose dissolution capacity can be increased at lower processing temperature. Importantly, this IL/cosolvent system reduce the quantity of IL used in cellulose dissolution [87]. Other recent studies also indicate the advantages of the mixed IL/cosolvent system for cellulose dissolution that time, temperature, and viscosity of cellulose mixture in IL solutions are reduced by the inclusion of a cosolvent [88,89]. Hawkins et al. [88] studied the dissolution time of flax fibers in EmimAc as a function of temperature and DMSO concentration. The dissolution rate was maximized when using an equal amount of EmimAc to DMSO, whereby the dissolution was about three times faster than that in the pure IL. The independence of an activation energy (E_a_), required for the dissolution of flax fibers in EmimAc, on DMSO concentration suggests that flax fibers dissolution is not primarily a viscosity-driven process. The result is related to the work of Chen et al. [90]; the dissolution is not only governed by viscosity, but also by solvent power.

Due to its low cost, low toxicity, and low viscosity, DMSO is a suitable candidate for a cosolvent. A series of IL/DMSO systems have been studied as shown in Table 1. The efficiency of IL/DMSO mixtures in cellulose dissolution and derivatization depends on the structures of ILs [91] and temperature [92]. With increasing temperature, the specific interaction between cellulose and the solvent increases [92]. Ferreira et al. [91] studied the effect of the presence of an ether linkage in the sidechain of imidazolium acetate (ImAc) using RMe_n_ImAc, R = 1-butyl or 2-methoxyethyl, *n* = 1 or 2 and the mixtures with DMSO. The mixture of BmimAc–DMSO was more efficient for the dissolution of MCC and acylation of cellulose because the mixture was less viscous, more basic, and formed stronger hydrogen bonds with cellobiose. Using the corresponding ILs with C2–CH_3_ instead of C2–H (1-butyl-2,3-dimethylimidazolium acetate and 1-(2-methoxyethyl)-2,3-dimethylimidazolium acetate) increased the concentration of dissolved cellulose [91]. Increasing the amount of DMSO does not only improve cellulose solubilization, but also leads to the formation of a more pronounced macroporous structure of the hydrogel [93].

In designing a new IL solvent system for cellulose dissolution, reversible/switchable reactions with CO_2_, a well-recognized greenhouse gas, have been extended [100,101,102]. Xie et al. [100] reported that the CO_2_-based reversible IL/DMSO solvent could dissolve MCC up to 10% *w*/*w* under mild conditions at 60 °C and a CO_2_ pressure of 2.0 MPa for 2 h, when 1,1,3,3-tetramethyl guanidine (TMG) was used as a base, in conjunction with methanol, ethanol, propanol, butanol, and ethylene glycol (Figure 2A). Yang et al. [103] also reported that DBU/DMSO could promote the dissolution of cellulose pulp at the relative low temperature of 50 °C in the presence of CO_2_. This is because the addition of CO_2_ causes the reversibility or switch-ability of this solvent system by changing the polarity from nonpolar to polar and reversing to its initial nonpolar state upon the release of CO_2_ [104,105]. Cellulose dissolves in this solvent system by nonderivative and derivative approaches (Figure 2B). In the nonderivative approach, simple alcohols such as methanol, hexanol, or ethylene glycol, in the presence of strong organic bases such as DBU, react with CO_2_ leading to the formation a DMSO–carbonate species solvent that can solubilize cellulose [100]. In the derivative approach, the superbase activates cellulose first and, thereafter, reacts with CO_2_ to form a DMSO-soluble cellulose carbonate intermediate [106]. 

Since the reversible reaction between cellulose, organic base, and CO_2_ is exothermic, the resulting high temperature is not suitable for cellulose dissolution. Therefore, the temperature used in most studies is in the range of 30–60 °C. At 30 °C, increasing the pressure of CO_2_ above 20 bar (2 MPa) increases in the carbonate formation and cellulose solubilization kinetics in 10 min, whereas at 40 bar (4 MPa), the carbonate formation is reached within 5 min. On the other hand, for CO_2_ pressures below 2 MPa, the solubilization of 3% *w*/*w* MCC is slower but occurs within 15 min [107]. At 30 °C and CO_2_ pressures of 0.2 MPa, the solubility of MCC in DMSO/DBU at weight ratio of DBU of 0.1 can reach 9% *w*/*w*. However, increasing the weight ratio of DBU to 0.5 slightly inhibits the MCC solubility. This indicates that the DBU amount influences MCC dissolution behavior [108]. At the higher temperature range of 40–60 °C, the dissolution of cellulose derived from a cassava pulp waste in DBU and ethylene glycol, which produces a switchable polarity solvent containing the cation of DBU (DBUH^+^) and the anion of ethylenedicarbonate (O_2_COCH_2_CH_2_OCO_2_^2−^) in the mixture with CO_2_, can be achieved in less than 30 min under CO_2_ pressure of more than 5.0 MPa [109]. Under 1 atm (0.1 MPa) in an open system, the obtained cellulose is stable and suitable for the preparation of cellulose hydrogel fiber by wet spinning [110]. Besides TMG and DBU, clear cellulose solutions were obtained in 1,5,7-Triazabicyclo [4.4.0] dec-5-ene (TBD) under 1 MPa CO_2_ at 50 °C for 3 h [111], and 2-tert-butyl-1,1,3,3-tetramethylguanidine (BTMG) under 1 atm (0.1 MPa) CO_2_ at room temperature for 5 min [112], but cellulose was not dissolved in triethyl amine (TEA) [111]. Instead of DMSO, other organic solvents with low Henry’s constants, such as propylene carbonate and sulfolane, are used as CO_2_ absorbents to increase CO_2_ absorption capacity in the solvent, so that the cellulose dissolution is enhanced. With the addition of propylene carbonate, the paper cellulose dissolution can be increased from 1.0 to 5.0% *w*/*w*, while those for microcrystalline cellulose and corncob cellulose are increased from 5.0 to 10.0 and 7.0% *w*/*w*, respectively, at 60 °C and a CO_2_ pressure of 0.5 MPa. The chemical and crystalline structures of the regenerated cellulose is not affected by the addition of these CO_2_ absorbents [113].

## 3. Regeneration of Cellulose in Antisolvents

The reorganization of the hydrogen bond network during the regeneration is a crucial step to obtaining the desired hydrogel properties. Cellulose regeneration occurs through a physical sol–gel process by using polar protic antisolvents, including ethanol, methanol, and water, that donate and accept hydrogen bonds to disrupt the favorable cellulose–IL interactions. Concomitantly, strong cellulose–cellulose interactions are promoted. The anion of the ILs affects the coagulation kinetics in the cellulose regeneration process. For example, in the case of ILs with Ac− anion, the addition of water as an antisolvent triggers the hydrogen bond destruction between cellulose and Ac− anion, and simultaneously induces the formation of hydrogen bonds between cellulose and cellulose, while Ac− forms a hydrogen bond with water. After the cellulose chains aggregate, the gelation is then induced, resulting in the precipitation of cellulose from IL/protic solvent mixtures [114,115]. This gelation occurs without a chemical crosslinker, because of the high entanglement density of cellulose [116]. The gelation of cellulose to hydrogels can be performed in different shapes (Figure 3).

While the mechanism of cellulose dissolution has been systematically investigated [117], cellulose regeneration phenomenon is inherently difficult to study. Zhou et al. [118] investigated the effect of different antisolvents on the regeneration from MCC/BmimCl solution using ATR–FTIR. The H–bond interaction between the chlorine ion and antisolvents is the driving force for cellulose regeneration. The crystallinity of the regenerated cellulose related to the H–bond acidity and basicity of the antisolvents [119,120]. For the phase separation of IL and cellulose in regeneration using water as the antisolvent from a MCC/EmimAc solution, a two-step model, involving the destruction and reconstruction of H–bonds based on the time-dependent FTIR spectroscopy, was proposed by Liu et al. [121]. In the first step, skeletons of the cellulose gel are built after the double diffusion of free IL and water, inducing structural change in cellulose regeneration. In the second step, the interaction between cellulose and IL is destroyed as water diffuses. The H–bond between the released anion of EmimAc and water is constructed, and the H–bond network among the cellulose hydroxyl groups is then reconstructed [121]. Relying on the double diffusion of antisolvents, a surfactant-induced and diffusion-driven method can be used to fabricate hierarchical porous cellulose hydrogel with permeability [122]. The mechanism of cellulose regeneration is also investigated by theoretical calculation and molecular dynamics (MD) simulation [123,124,125]. Fu et al. [126] studied the mechanism of cellulose regeneration from theoretical calculation by analyzing the structures and hydrogen bonds of BmimAc-nH_2_O and cellobiose-ILs-nH_2_O (*n* = 0–6) clusters. When numbers of water molecules (*n*) were ≥5 in the BmimAc-nH_2_O clusters, the solvent-separated ion pairs played a dominant role in the system. The cation and anion interaction could be separated by water in reducing cellulose dissolution by ILs, when numbers of water molecules were increased. Furthermore, a more stable structure with high hydration in an aqueous solution of BmimAc-nH_2_O and cellobiose-ILs (*n* = 0–6) clusters was found. In the presence of water molecules in the system, the hydrogen bonds among H_2_O, the hydroxyl group of cellulose, and the oxygen of acetate were formed. Therefore, cellulose regeneration was promoted by the decrease of the interactions between cellulose and ILs [126]. After the dissolution and regeneration process, the transition from cellulose I to II occurs [127,128]. In the CO_2_/DBU/DMSO solvation system, cellulose that is regenerated through integrated thermally induced CO_2_ release is cellulose IV_I_ crystalline structure [102].

**Figure 3 gels-09-00546-f003:**
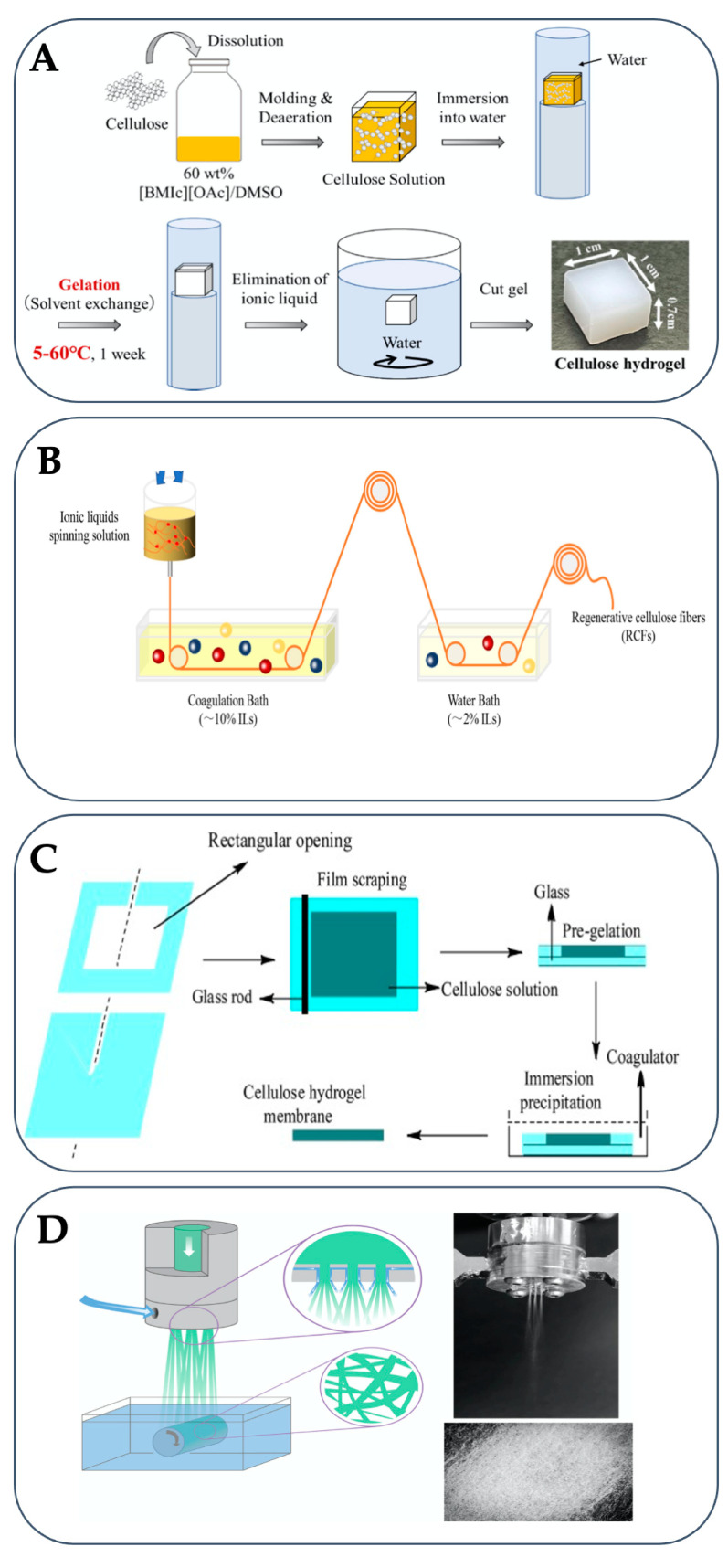
Ionic liquid-based preparation processes of cellulose hydrogels in the form of bulk (**A**) (Reprinted with permission from ref. [95] Copyright © 2023, Elsevier, B.V.), fiber (**B**) (Reprinted with permission from ref. [129] Copyright © 2017 Wiley Periodicals, Inc.), membrane (**C**) (Reprinted with permission from ref. [130] Copyright © 2017 Wiley Periodicals, Inc.), and nonwoven mesh (**D**) (Reprinted with permission from ref. [131] Copyright © 2019 Wiley Periodicals, Inc.).

The characteristics of the final hydrogel are related to the regeneration technique and composition of the regeneration bath [132]. The density of the regenerated cellulose hydrogel depends on the type of antisolvent that determines the mixing rate between the IL and the antisolvent during the regeneration process [104] (Figure 4A). The reduced regeneration rate of cellulose chains by the low mixing rate results in an ordered structure and a transparent cellulose hydrogel. Hence, mechanical characteristics are also affected [133,134]. Therefore, finding a suitable antisolvent is a key parameter. Water is mostly used due to the most environmentally friendly and inexpensive antisolvent. The regenerated cellulose using water had a higher crystallinity and better thermal stability than that using ethanol [135]. Due to the hydrotropic nature of the more bulky cations of IL, cellulose regeneration may require more hydrophobic mixtures (e.g., water–alcohols) for a more efficient regeneration of pure cellulose [136]. The reconstitution of cellulose molecules with various antisolvents affecting the multiscale structures and properties of cellulose has been studied. Gao et al. [137] employed different coagulation baths of ethanol, methanol, NaOH, and H_2_SO_4_ aqueous solutions to prepared cellulose hydrogel from TMGH OOCOCH_2_CH_2_OCOO/DMSO (the mole fraction of IL in the mixed solvent is 0.2). The hydrogel regenerated from alcohol had a more amorphous crystal structure, a smoother surface morphology, and higher transparency properties than those regenerated from 5% *w*/*w* NaOH and 5% *w*/*w* H_2_SO_4_ aqueous solutions (Figure 4B). The hydrogel from ethanol exhibited good thermostability and mechanical properties (tensile strength and strain of 56.2 MPa and 20.4%, respectively, and a Young’s modulus of 901 MPa). Water vapor permeability of 8.7 × 10^−3^ g μm/m^2^ day kPa and oxygen permeability of about 4.0 cm^3^ μm/m^2^ day atm were obtained. The hydrogels regenerated from aqueous solutions of alkali and the acid-coagulating baths shielded oxygen in order to permeate [137]. In the case of regenerating cellulose from an EmimDEP solution, the regenerated cellulose fiber hydrogel shows differences in its internal structure and mechanical properties due to the different diffusion rates between the IL solution and coagulation bath containing 10% *w*/*w* of IL and the antisolvent (Figure 4C). In comparison with the pure water, the absolute ethanol slightly reduces the crystallinity and orientation, whereas the elongation slightly increases [138]. The coagulating bath containing IL has consequences for an up-scaled continuous process, such as the possibility to regenerate cellulose in an IL-concentrated bath. This minimizes solvent recycling costs using, for example, the thermal separation of IL and water. A closed-loop process operation, performing single filament spinning in IL/H_2_O mixtures, was simulated and studied by Guizani et al. [139]. Spinnability and preserved mechanical properties of the hydrogel fibers of cellulose dissolved in DBNHAc were achieved in a coagulation bath containing up to 30% *w*/*w* IL, although the fibrillar structure was less pronounced in comparison to a pure water bath. At 45 % of IL in the bath, the spinnability was possible but resulted in lower quality fibers because the mechanical properties of the hydrogel fibers started to deteriorate [139]. 

A switchable coagulant is also used to regenerate cellulose from the IL solution, and it is easier to remove from the IL [140,141]. This approach can minimize the direct evaporation process in the recovery of hydrophilic ILs from an aqueous solution, which is energy consuming. The CO_2_ is used as a coagulation solvent for the separation of cellulose and IL [140]. Since CO_2_ is miscible in the IL, such as BmimAc, but immiscible with cellulose, after depressurization and removal of CO_2_, the cellulose precipitated as a transparent gel [142]. The yield of the regenerated cellulose (60%), prepared from 10% *w*/*w* MCC in BmimAc, increases after 3 h, under 6.6 MPa compressed CO_2_, and at 25 °C [143]. However, more cellulose can be regenerated from the IL/cosolvent systems than this neat IL system [108,143]. The degree of polymerization (DP) value of the regenerated cellulose under the neat system decreases from 282 of MCC to 202, depending on the reaction time of the compressed CO_2_, as well as the pressure and temperature. The DP is higher than those regenerated by a cosolvent system (DP = 157–175) and ethanol antisolvent (DP = 163) [143], but slightly lower than those regenerated using water (DP = 210) [144]. Upon the reaction of amine with CO_2_, the non-ionic amine is changed to ionic carbamate, which can be employed in the formation of switchable solvents [145]. These reactions of amines with CO_2_ in the field of CO_2_ capture and storage have been studied recently [146,147]. The CO_2_ capture occurs by the formation of an insoluble monoethanolamine–carbamate, 2-hydroxyethylammonium (2-hydroxyethyl) carbamate. By heating and/or reducing the pressure, the captured CO_2_ is released from the carbamate [141]. This approach to coagulate cellulose in an IL solution, by the formation of a carbamate salt having strong hydrogen bond donors and acceptors, can disrupt the hydrogen bonding between cellulose and the IL anion, resulting in coagulation. The reversible coagulant approach also facilitates the recovery of the IL to avoid using the commonly used liquid antisolvent, thus saving on the cost and energy necessary for liquid evaporation [141].

## 4. Applications

### 4.1. Electrochemical and Electronic Materials

Active research on electrochemical energy storage has been stimulated due to the increasing need for energy storage devices. Depending on the balance between energy and power density, electrochemical energy storage is categorized into batteries and supercapacitors. These devices possess a long cycling life and instantaneously store and release energy, allowing the potential for high efficiency energy storage devices [148,149]. The more environmentally friendly production of energy storage device technologies is a matter of interest in the use of biopolymer-based hydrogel electrolytes [150]. For instance, cellulose-based supercapacitors have gained attention due to their mechanical strength, environmental friendliness, and light weight [151,152,153]. Low-cost cellulose also decreases the total electrode cost, which is beneficial for producing a competitive electrode material for electricity storage on a large scale [154]. 

The addition of graphene allows the cellulose hydrogel to improve the thermal and electrical conductivity suitable for energy storage applications, whereas the cellulose hydrogel provides graphene at a low-cost template for large-scale applications. Graphene, a one-atom-thick two-dimensional sp^2^-bonded carbon material, has high thermal conductivity (~2000–4000 W/m/K) and mechanical properties. It is a promising filler to improve the thermally conductive and mechanical properties of polymer matrices, which has been frequently reported in recent years [155,156,157,158,159]. Xu et al. [160] reported the use of wood pulp cellulose as the raw material to prepare cellulose/graphene composite hydrogels by dissolving in BmimCl as the medium to dissolve cellulose and disperse graphene, and then coagulating the hydrogel in water. Enhanced mechanical strength and thermal stability was observed in comparison with pure cellulose hydrogels. Their appearances and mechanical properties are shown in Figure 5A–D. With 0.5% *w*/*w* of graphite oxide in cellulose, about four times higher compressive Young’s modulus was obtained due to an organized alignment of cellulose chains on graphene sheets and an interfacial adhesion between graphene and cellulose chains. However, at the higher doping, more graphene led to enhanced mechanical properties of the hydrogel because the uniformly dispersed graphene/cellulose had a typical 3D network structure. This made the cellulose/graphene composite hydrogels become more compact. It is also reported that the mechanical strength of the composite hydrogel via π–π stacking interactions is much higher than that of the pure cellulose hydrogel via hydrogen bonds [160]. The graphene is functionalized with IL by cation–π or π–π and electrostatic interactions [161]. The technique containing graphene and IL can improve the performance of the corresponding composite hydrogel in the wider applications of supercapacitors [148], electrocatalysts [162], and sensors [163]. 

The aqueous phase in hydrogels, and the liquid-like transport properties of the hydrogels, enable their fast diffusion for conductivity [166]. When the hydrogels are treated with graphene, or other conductive materials such as polypyrene, they can transfer electrons [164]. Liang et al. [164] dissolved MCC in BmimCl solution and crosslinked by N,N′-methylene bisacrylamide to obtain cellulose hydrogel. The hydrogel was then immersed in a mixed aqueous solution of ferric trichloride and sodium p-toluenesulfonate, and in the pyrrole solution, to form a conductive hydrogel. The mechanical strength was also improved in comparison to the typical cellulose hydrogel (Figure 5E,F). Such ionic conductive hydrogels, prepared by immersion/swelling in electrolyte solutions, have also attracted an increasing interest in terms of batteries [165,167] and solar cells [168]. The cellulose hydrogel has properties superior to some commercial products, such as transparency, thermal stability, and ionic conductivity (Figure 5G–J). 

In modern electronics, for instance, in human wearable sensors and sensors in soft robotics, in the development of soft electrolytes with high stretchability, compressibility, and ion transport are in demand [169,170]. As triboelectric nanogenerators, mechanical energies in human activities can be turned into electricity because of the functions of stretchability, transparency, ionic conductivity, and even self-healing behaviors [171,172]. This also make them suitable for wearable electronics [173]. Various polymer systems, such as PAA/PDMAPS/IL hydrogel as triboelectric nanogenerator [174], HPC/PVC/NaCl hydrogel as an artificial nerve allowing the passing of stable electrical signals [175], PVA/PAM/NaCl hydrogel as a high-sensitivity strain sensor [176], cellulose nanofibers/PVA/AlCl_3_ hydrogel as artificial electronic skins [177], and cellulose nanofibers/PAM–AA/LiCl hydrogel as a muscle-like sensor for human motion monitoring [178], have been reported. For these applications, the use of hydrogels has two drawbacks, which are water retention ability and narrow temperature range [179,180]. As water evaporates from a hydrogel, the ionic conductivity of the hydrogel decreases due to the evaporation of water at high temperatures [181]. At subzero temperatures, the hydrogels become rigid and restrict the ionic transport [182]. Hence, hydrogels are not suitable for long-term applications in dry, subzero or hot environments. Ionogels containing IL can be used as electrodes in electronic devices at high temperatures or in dry environments due to their nonflammability and negligible vapor pressure [174]. The ionogels can be prepared from various types of cellulose, e.g., micro and non-crystalline, bacterial cellulose, and cellulose derivatives [183]. Cellulose ionogels are a promising solid electrolyte due to their high conductivity (0.01–8 S/m), wide electrochemical potential window, and thermal stability up to 200 °C [167,184,185]. Since the properties of the ionogel are related to the properties of the IL, imidazolium cations such as Amim^+^ and Bmim^+^, having high electrical conductivity and the ability to dissolve cellulose and to form gel for electrochemical applications, are the focus of most studies [185,186,187,188]. Zhao et al. [188] developed a topology-tunable dynamic hydrogel containing cellulose, BmimCl, and water. The prepared gel exhibited tunable properties of mechanical strength, ionic conductivity, viscoelasticity, and self-healing. With 32% *w*/*w* of water, the microstructure switched to a dense and compact network, giving the gel good stretchability, robust toughness, a high ionic conductivity, transparency, and biocompatibility. This demonstrated that this hydrogel had a potential for use in electronic skins and intelligent devices [188]. 

### 4.2. Membranes for Water Treatments

In membrane technology for water and wastewater treatments, cellulose-based membranes are well known [189]. Cellulose composite nanofiltration membranes show high rejection performance for the separation of different dyes. Anokhina et al. [190] studied the filtration of two anionic dyes, Orange II (350 g/mol) and Remazol Brilliant Blue R (626 g/mol) by applying a cellulose membrane prepared using IL as a membrane casting solution. The high concentration of cellulose in IL led to the formation of a denser cellulose-based selective layer in the membrane. In addition, the separation selectivity indicated by rejection factors of the two dyes (R_350_ and R_626_, respectively) varied during filtration through the membranes prepared from an 8% *w*/*w* cellulose solution in varying EmimAc and DMSO ratios. At 90% *w*/*w* EmimAc in DMSO, R_350_ and R_626_ were 40% and 65%, respectively. The rejection factors of the two dyes increased with an increase in cellulose concentration in the casting solution due to the formation of a denser porous structure of the cellulose layer. At 16% *w*/*w* cellulose in 50% *w*/*w* EmimAc in DMSO, R_350_ and R_626_ were 61 and 82%, respectively [190], whereas, in the system of 14% *w*/*w* cellulose in pure EmimAc, R_350_ and R_626_ were 24 and 31%, respectively [96]. In the diluted IL system (14% *w*/*w* cellulose in 50% *w*/*w* EmimAc in DMSO), replacement of water with alcohols as antisolvents in regeneration step can alter the membrane permeability and rejection of Orange II and Remazol Brilliant Blue R [96]. Ilyin et al. [96] reported that methanol led to a fivefold increase in membrane permeability of DMF as compared to that of the membrane regenerated in water (P_DMF_ = 0.23 kg/m^2^/h/atm), but decreased its rejection to 35–47% due to an increase in the porosity of the membrane. Ethanol yielded a membrane with lower permeability, but better rejection (56–71%). As compared to methanol, ethanol had a lower diffusion rate, leading to the contraction of the forming film, reducing its pore size. Isopropanol provided an even lower diffusion rate and delayed structural changes of membrane during phase separation and eliminated the effects of network contraction. However, the film had better porosity and permeability compared to that regenerated by ethanol [96]. The dense layer of the membrane and the size exclusion of the dye are the two main factors of dye rejection [191]. Solute diffusion is the transport mechanism through the cellulose-based membranes. In addition, the surface charge and hydrophilicity are the dominant factors of the antifouling capability of the membranes [191]. Esfahani et al. [191] studied 10% *w*/*w* bamboo cellulose-based membranes and prepared them using BmimCl and coagulated them in water, showing the high rejection (>80%) of crystal violet and 60–70% for methylene blue and methylene orange at a dye concentration of 10 ppm in water. This is because of the dense layer and hydrophilicity of the membrane (Figure 6A–C). The membrane prepared from 3% bamboo cellulose showed low rejection (35%) of the three dyes, but high water flux of 600 L/m^2^/h, respectively. The dye separation efficiency of the cellulose membranes also depends on pH, temperature, ionic charge and concentration, and reuse cycle (Figure 6D). Comparing ~2–3% *w*/*w* lemon peel cellulose [192] and pineapple peel cellulose [193] prepared in BmimCl and coagulated in water, the adsorption capacities of methylene blue are 25 and 57 mg/g from the initial dye concentration of 50 and 100 ppm, respectively.

### 4.3. Bioink for 3D Printing

Three-dimensional (3D) printing, known as rapid prototyping technology (RP) [194], additive manufacturing technology (AM) [195], or solid freeform fabrication (SFF) [196], is the technique for constructing solid objects by depositing several layers of material in sequence. Due to the capability of fabricating cost-effective and customizable objects, 3D printing has been recognized as a promising technology [195]. With development in science and technology, it is sufficiently mature to be applied to lower cost materials by open-source software [197]. To date, this technology has been applied in the medical [198], pharmaceutical [199], education [200], clothing [201], and food [202] industries. These 3D-printed objects have proven to have effective applications in cell culture and drug delivery [203,204,205]. One of the primary reasons for using biopolymers in 3D printing is the dispersion stability and flow behavior of inks, as well as structural and thermal properties of the printed objects [206,207]. The desired mechanical properties of 3D-printed constructions are conferred by inducing self-assembled structures and strengthening the interlayer bonding [206]. There has been an increase in the number of studies using cellulose or cellulose combined with other biopolymers in 3D printing [208,209,210,211]. The alignment of CNC and other anisotropic particles during 3D printing mimics the multiscale architectures of biological materials such as wood, plant stems, bone, and mollusk shells [212]. Such particle alignment by shear stresses (shear-inducing flow) is intrinsically present in extrusion-based 3D printing processes [210,213]. However, the concentration of CNC that can be added to the inks remains very limited [214], which is appropriate for the printing of cellulose micropatterns on flat substrates [215] or the printing of liquid resins into CNC-laden complex geometries [216,217]. It is not currently feasible to print 3D objects at high cellulose content without nozzle clogging [218]. The content of cellulose nanofiber that can be used in the ink is even lower than that of CNC, due to the high amount of surface hydroxyl groups and entanglements via hydrogen bonding between the fibers leading to an increased yield stress and the high pressure needed for the extrusion of the ink [219,220]. The ILs allow higher concentrations of cellulose in inks compared to water-based systems [221]. 

Markstedt et al. [221] showed the possibility of printing membrane or spatially tailored 3D hydrogels by using cellulose dissolved in EmimAc, followed by coagulation in water. The hollow cylindrical structures were created by complex patterns of 2D structures and multilayered printing constructs. High cellulose concentrations and molecular weight were shear-thinning and favorable for printing. A solution of 4% *w*/*w* cellulose, dispensed at the pressure for the extrusion of 6 bars with a 12.7 mm long needle, having an inner diameter of 0.41 mm at a flow rate of 10 μL/min, enabled the prints to obtain an acceptable resolution. The rates of coagulation had to be considered carefully in the material construction and selection of printing parameters. If the coagulation occurred too quickly it would result in poor adhesion between printed layers and collapse, whilst coagulation that was too slow led to height limitations [221]. By using 4% *w*/*w* bleached kraft pulp cellulose dissolved in EmimDMP, obtaining a viscosity of 740 Pa·s and a yield stress of 2.5 kPa, Hopson et al. [222] reported that the printed ionogels and the hydrogels had a good print resolution and fidelity due to the low volatility of IL. The results were comparable to the hydrogel prepared by the commercial ink (Cellink Xplore^TM^) [222]. Because of their increased viscosity and yield stress, caused by the high cellulose content and viscose IL in inks, cosolvents, such as 1-butanol and DMSO, can modulate the printability [223]. For instance, DMSO in the range of 41–47% *w*/*w* with EmimAc or BmimAc yields ink within the printable range for inkjet printing of 1.0–4.8% *w*/*w* cellulose (printability parameter <10 at 55 °C). By using water as an antisolvent, the regenerated ink hydrogel on a PET substrate exhibits structural distortion, whereas there is no such issues on glass [223]. Using 30–40% *w*/*w* water as a cosolvent with bacterial cellulose dissolved in choline acrylate (a polymerizable IL) was studied by Fedotovam et al. [224,225]. The addition of water exhibited the printability and stability of inks. The solvent viscosity and yield stress were decreased from 2230 Pa (anhydrous system) to 97–160 Pa. The decreased viscosity of the dispersion caused by the water–acrylate complexes led to the destruction of the network of ionic bonds between IL cations and anions. However, cellulose in this binary solvent was nearly a suspension of the cellulose nanofibers, not a solution. The MD simulations provided information that water changed the structural organization of the hydration of acrylic ions at 30–40% of water. The IL/water ratio for 3D printability related to the ion–dipole interactions between IL components and between cellulose–IL in the ink, which decreased with the addition of water. This IL/water ratio was enough for the stabilization of the dispersion, while the morphology and crystalline structure of the cellulose nanofibers were preserved. When further increasing the water content (50% *w*/*w*), the yield stress increased due to the amorphization, swelling, and agglomeration of cellulose nanofibers [224,225].

## 5. Conclusions and Outlook 

This review summarizes the recent studies that examined the preparation of cellulose hydrogels and related products by dissolving cellulose in ILs and then regenerating it in an antisolvent such as water or alcohol. Overall, these studies indicate that the properties of the hydrogels can be tuned through varying the concentrations of cellulose in the solution, the type of ILs/cosolvents, and the compositions of antisolvents in the regeneration process. Imidazolium ILs, paired with acetate and chlorine, have been widely used in cellulose dissolution. However, the development of new ILs for higher cellulose dissolution are preferable, where dissolving at room temperature, the lower viscosity of cellulose solution, the low decomposition/depolymerization of cellulose, and low cost, have been challenges. The cosolvents, such as polar aprotic organic solvents, have been mixed with the recently developed imidazolium ILs to achieve this goal. The CO_2_ switchable solvent system for cellulose dissolution has been explored as a new platform for cellulose processing. The dissolution of cellulose is efficient, even though it is performed at room temperature. Importantly, low IL content has been shown to be effective. This can address the remaining problem of using IL in terms of cost, since ILs are more expensive than other solvents. However, cellulose solubilization conditions in this CO_2_ solvent need to be optimized since the cellulose dissolution depends on cellulose source, type/amount of the superbases, pressure of CO_2_, temperature, and time. In addition, the preparation of cellulose hydrogels in their various forms, such as membranes and fibers regenerated from cellulose dissolved in these CO_2_ solvents, has not systematically and widely studied. Therefore, it is of great interest to study the conditions of the regeneration process, such as composition, temperature of the coagulation bath, and time to reach the gel state. 

The recycling and purification of ILs are essential for an economical and sustainable cellulose hydrogel manufacturing process. Water or other antisolvents added to regenerate cellulose need to be efficiently separated from the ILs. Several methods have been reported, from simple evaporation to more complex phase separation processes, but most studies in this field are still preliminary. In-depth knowledge is essential for improving the recyclability and long-term chemical stability of the ILs. The separation process should be efficient, easy to operate under mild conditions, and energy-saving, which are beneficial for the scaling-up of the manufacturing process. Due to the capability of recycling and purification, the cellulose-based hydrogels, prepared by using the IL solvent, will truly follow the 17 Sustainable Development Goals (SDGs) outlined in the resolution of the United Nations that highlight the need for green and sustainable solvents.

## Figures and Tables

**Figure 1 gels-09-00546-f001:**
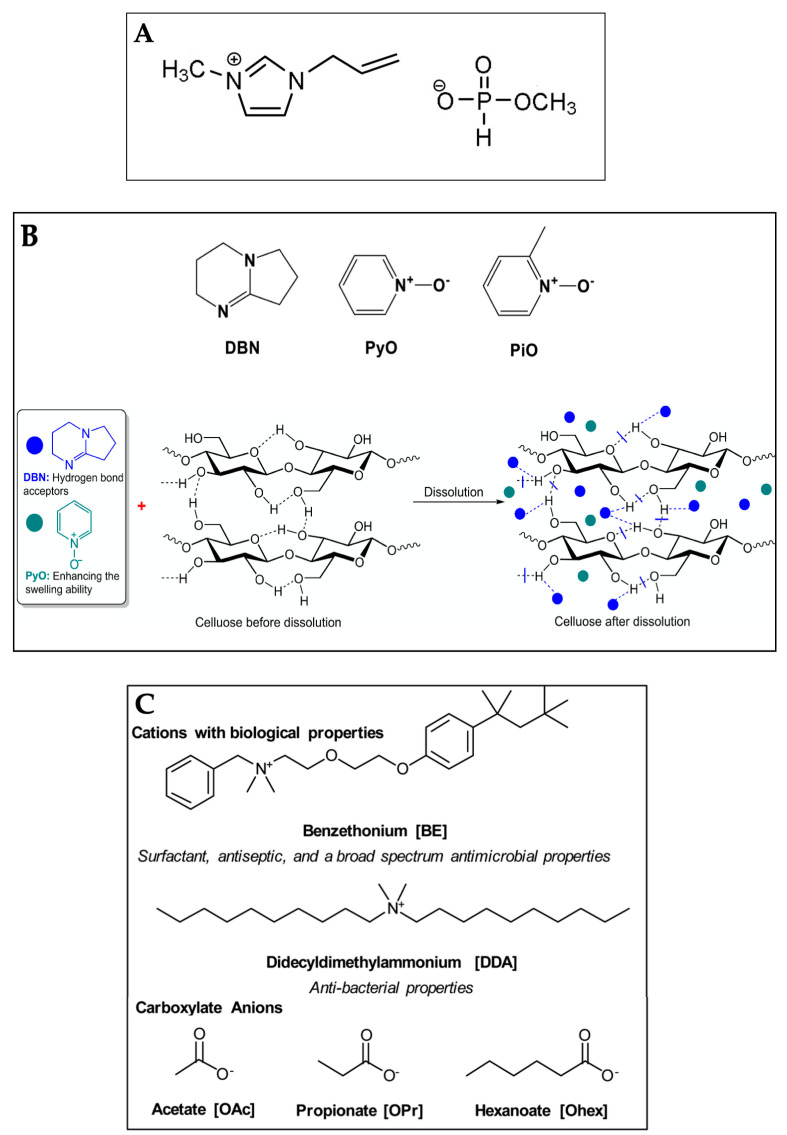
Chemical structures of 1-allyl-3-methylimidazolium methyl phosphonate (Amim (MeO)PHO_2_) (**A**) (Reprinted with permission from ref. [76] Copyright © 2019, Elsevier, B.V.), 1,5-diazabicyclo [4.3.0] non-5-ene (DBN), pyridine N-oxide (PyO) and 2-picoline N-oxide (PiO) with graphic dissolution mechanism of cellulose (**B**) (Reprinted with permission from ref. [82] Copyright © 2021, Elsevier, B.V.), and benzethonium (BE)- and didecyldimethylammonium (DDA)-based ILs combined with short alkyl carboxylate anions (**C**) (Reprinted with permission from ref. [83] Copyright © 2021, Royal Society of Chemistry).

**Figure 2 gels-09-00546-f002:**
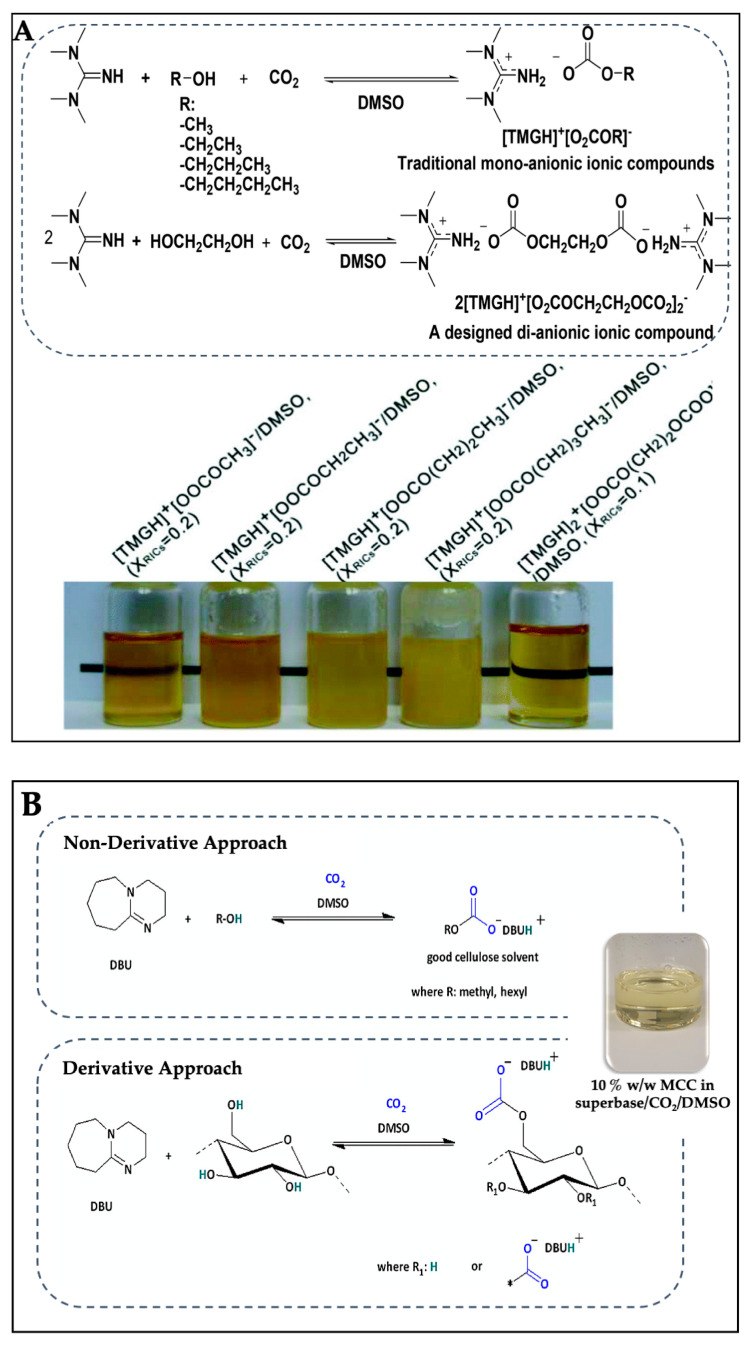
Reaction of 1,1,3,3-tetramethyl guanidine (TMG) with methanol, ethanol, propanol, butanol, and ethylene glycol in the presence of CO_2_ and dimethyl sulfoxide (DMSO) to form reversible ILs (TMGHO_2_COR) (**A**). Pictures illustrate 5% *w*/*w* of microcrystalline cellulose (MCC) solution in the reversible ionic compound (RIC) solvents at the different molar fractions in DMSO after CO_2_ capture (X_RIC_). (Reprinted with permission from ref. [100] Copyright © 2014, Royal Society of Chemistry). Nonderivative and derivative approaches of the CO_2_ switchable 1,8-diazabicyclo [5.4.0] undec-7-ene (DBU) super base solvent system (**B**). Picture illustrates 10% *w*/*w* of MCC solution in superbase (DBU)/CO_2_/DMSO solvent. (Reprinted with permission from ref. [104] Copyright © 2019, American Chemical Society).

**Figure 4 gels-09-00546-f004:**
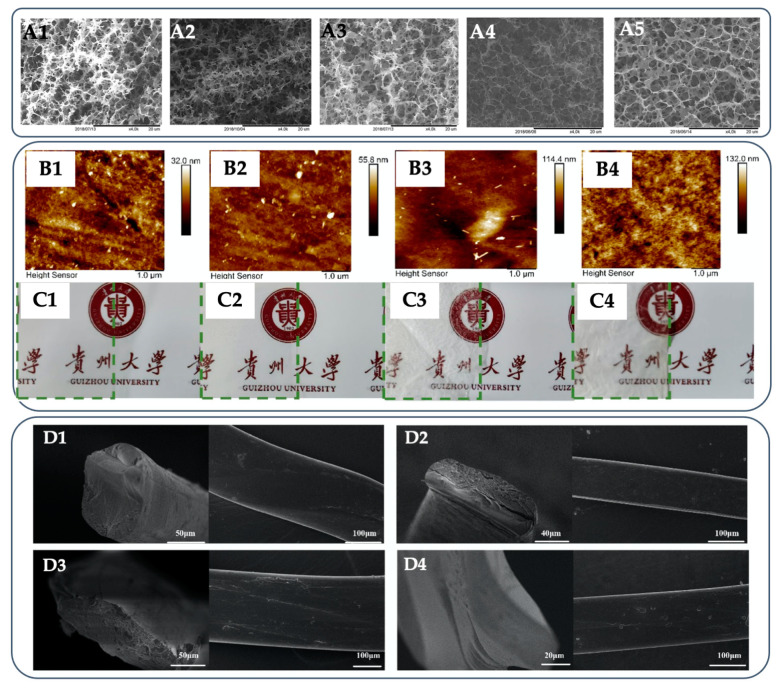
Morphologies of cellulose hydrogels coagulated in different compositions of the regeneration bath. SEM images of cellulose hydrogels prepared from 5% *w*/*w* microcrystalline cellulose (MCC) in 1,5-diazabicyclo [4.3.0] non-5-ene (DBN)–CO_2_ solvent system and coagulated using water, methanol, ethanol, isopropanol, and no coagulating solvent (**A1**–**A5**) (Reprinted with permission from ref. [104] Copyright © 2019, American Chemical Society). AFM images (**B1**–**B4**) and photographs (**C1**–**C4**) showing transparency of cellulose hydrogels prepared from 5% *w*/*w* MCC in 1,1,3,3-tetramethyl guanidine (TMG)/dimethyl sulfoxide (DMSO) solvent system and coagulated using ethanol (**B1**,**C1**), methanol (**B2**,**C2**), 5% *w*/*w* NaOH (**B3**,**C3**), and 5% *w*/*w* H_2_SO_4_ (**B4**,**C4**) aqueous solutions (Reprinted with permission from ref. [137] Copyright © 2021, American Chemical Society). SEM images of cellulose hydrogels prepared from 12% *w*/*w* in 1-ethyl-3-methylimidazolium diethyl phosphate (EmimDEP) and coagulated in water (**D1**), ethanol (**D2**), and mixed coagulating solvents containing ethanol:water:IL of 7:2:1 (**D3**) and 1:8:1 (**D4**) (Reprinted with permission from ref. [138] Copyright © 2022, Elsevier, B.V.).

**Figure 5 gels-09-00546-f005:**
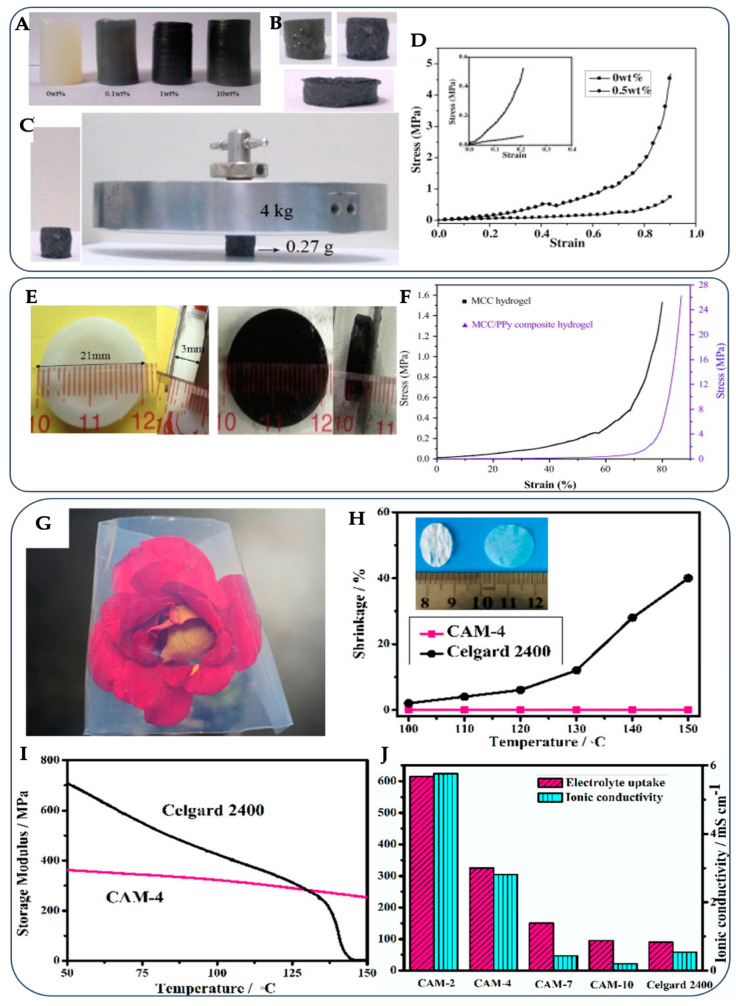
Appearance and properties of cellulose hydrogels prepared by using ILs, e.g., 1-buthyl-3-methyl imidazolium chloride (BmimCl) and 1-allyl-3-methyl imidazolium chloride (AmimCl), and an addition of graphene oxide (**A**–**D**), polypyrrole (PPy) (**E**,**F**), and liquid electrolyte (**G**–**J**). Images of composite cellulose hydrogels containing 5% *w*/*w* of cellulose and 0, 0.1, 1, and 10% *w*/*w* of graphene oxide before (**A**) and after (**B**) freeze-drying. The freeze-dried composite hydrogel supporting a counterpoise (**C**). Compressive stress–strain curves of the cellulose hydrogels containing 0 and 0.5% *w*/*w* of graphene oxide (**D**). (Reprinted with permission from ref. [160] Copyright © 2015, Elsevier, B.V.). Images (**E**) and compressive stress–strain curves (**F**) of the cellulose hydrogels containing 2.5% *w*/*w* of MCC and PPy. (Reprinted with permission from ref. [164] Copyright © 2014, Elsevier, B.V.). Image (**G**) and thermal properties (**H**,**I**) of cellulose hydrogel containing 4% *w*/*w* of cellulose (CAM-4). Electrolyte uptake and ionic conductivity of cellulose hydrogels containing 2–10% *w*/*w* of cellulose compared with those of Celgard 2400 commercial separator (**J**). (Reprinted with permission from ref. [165] Copyright © 2017, American Chemical Society).

**Figure 6 gels-09-00546-f006:**
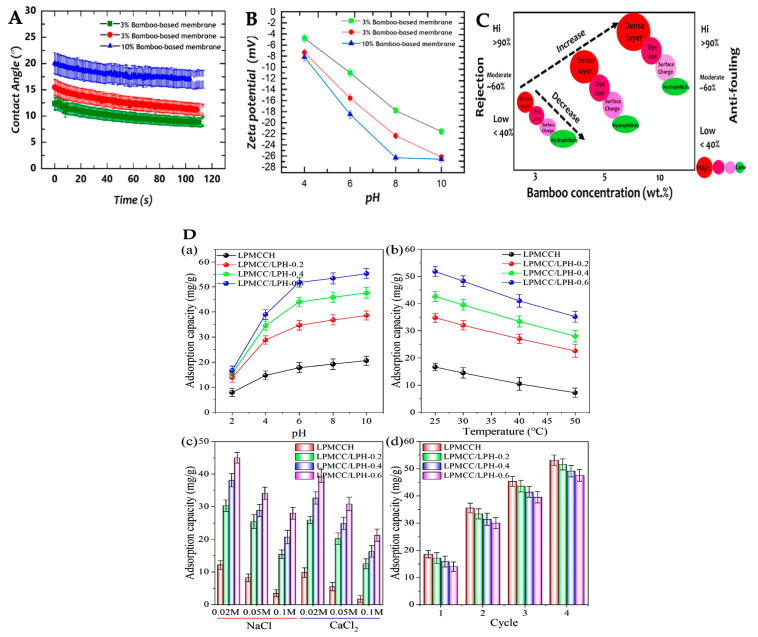
Properties and efficiencies cellulose membranes prepared by dissolving bamboo cellulose (**A**–**C**) and lemon peel cellulose mixed with raw lemon peel (**D**) in 1-buthyl-3-methyl imidazolium chloride (BmimCl) as the casting solution. Dynamic contact angle (**A**), zeta potential (**B**), and effectiveness (**C**) of membranes prepared from bamboo cellulose at the concentration of 3, 5, and 10% *w*/*w*. (Reprinted with permission from ref. [191] Copyright © 2020, American Chemical Society). Adsorption capacity of cellulose hydrogel (**D**) prepared by dissolving lemon peel cellulose without mixing raw lemon peel (LPMCCH) and with mixing the lemon peel of 0.2–0.6 g (LPMCC/LPH). Effects of pH (**a**), temperature (**b**), ionic type and strength (**c**), and adsorption/desorption cycles (**d**). (Reprinted with permission from ref. [192] Copyright © 2021, Elsevier, B.V.).

**Table 1 gels-09-00546-t001:** Recent studies on IL/DMSO solvent systems used for cellulose dissolution.

IL	Concentration of DMSO in IL and Temperature to Dissolve Cellulose	Concentration of Cellulose	Ref.
C_3_OMeImAc	60% mole, 60 °C	12% *w*/*w* MCC ^a^	[91]
C_4_MeImAc	60% mole, 60 °C	16% *w*/*w* MCC ^a^	
(BmimAc)			
C_3_OMe_2_ImAc	60% mole, 60 °C	19% *w*/*w* MCC ^a^	
C_4_Me_2_ImAc	60% mole, 60 °C	22% *w*/*w* MCC ^a^	
BmimAc	20% *w*/*w*, 70 °C	8% *w*/*w* MCC, Avicel, and α-cellulose ^b^	[93]
	40% *w*/*w*, R.T. ^c^	20% *w*/*w* Avicel ^b^	[94]
	40% *w*/*w*, R.T. ^c^	~10% *w*/*w* MCC ^b^	[95]
	75% *w*/*w*, 80 °C	14% *w*/*w* cellulose powder ^b^	[96]
BmimCl	50% *w*/*w*, 15–75 °C	3% *w*/*w* cellulose powder ^b^	[92]
	30% *w*/*w*, 120 °C	-	[97]
EmimAc	50% *w*/*w*, 60 °C	2.5% *w*/*w* cotton ^b^	[98]
	25% *w*/*w*, 50 °C	10% *w*/*w* MCC ^b^	[99]
	75% *w*/*w*, 80 °C	14% *w*/*w* cellulose powder ^b^	[96]
EmimCl	75% *w*/*w*, 80 °C	14% *w*/*w* cellulose powder ^b^	[96]

^a^ solubility test. ^b^ fixed amount. ^c^ room temperature (~25 °C).

## Data Availability

No new data were created.

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
