# Peer review of "Recent Advances in Cellulose-Based Hydrogels Prepared by Ionic Liquid-Based Processes"

_gels, 2023, doi:10.3390/gels9070546_

Round 1

Reviewer 1 Report

This study reviewed the recent advances in preparing cellulose hydrogels by dissolving cellulose in ILs and then regenerating it in antisolvents such as water and alcohols. Overall, these studies indicate that the properties of the hydrogels can be tuned through concentrations of cellulose in the solution, types of ILs and cosolvents, and compositions of antisolvents in the regeneration process. The manuscript is well written and prepared. However, some points need to be considered by the authors.

  1. Please include more information about your review content in the abstract.
  2. What advantages can this paper give readers? Please add it to the last sentence of the abstract.
  3. Each keyword should start with a capital letter.
  4. Highlight the novelty and significance of this study in the last paragraph of the introduction section.
  5. Lines 437, The font of this sentence is different.

Reviewer 2 Report

This review summarized the preparation of cellulose-based hydrogels by using ionic liquid (IL)-based processes. My suggestion is that the structure of the paper needs to be revised. The specific comments are as follows:

1. It is recommended to change the proportion of the images in the manuscript and pay attention to the layout of the images.

2. It is suggested that the authors may add a brief description of other cellulose dissolving systems.

3. The classification of the applications of cellulose materials prepared from ionic liquids is too few, and it is believed that there are more than the two directions mentioned in the applications, and that the summary is too simple.

4. The article spends a lot of space on IL dissolving cellulose and too little on introducing regenerative gels, so it is recommended to expand accordingly.

5. "Due to the reversible reaction among cellulose, organic base, and CO2 is exothermic, high temperature is not suitable for the cellulose dissolution.", Does the increased temperature of the CO2 exothermic process affect the dissociation and can it be controlled? Does it cause cellulose to break down?

The author is advised to polish the article.

Reviewer 3 Report

The review by Taokaew S. focuses on the preparation of cellulose hydrogels and related products via cellulose solutions in pure and dilute ionic liquids and their subsequent applications. The author comparatively discusses the dissolution mechanism and properties of cellulose solutions in pure ionic liquids and in mixtures of ionic liquids with co-solvents, then describes the process of regeneration of cellulose from its solutions with non-solvents, and finally considers possible applications of the precipitated products in energy storage and membrane technologies.

In summary, the review is well-written and easily readable and can be published after the following additions are made:

Line 40: “Cellulose hydrogels are a 3D network of crosslinked polymer molecules”. This is not the correct phrase. There are two types of cellulose hydrogels. The first is (as the author writes) chemically cross-linked cellulose macromolecules pre-dissolved or swollen in water. The second type is cellulose micro- or nanoparticles, which are dispersed in water and form coagulation contacts and a 3D network (see, e.g., 10.1007/s10570-018-1678-6). The cellulose macromolecules in the second type of gels from nano- or microfibrillated cellulose are not cross-linked (10.1016/j.indcrop.2016.02.016, 10.1007/s10570-010-9405-y).

Line 60: “N-methyl morpholine oxide (NMMO) [31,32]”. For a sol-gel process of cellulose, NMMO/DMSO mixtures were also used as a solvent (10.1007/s10570-021-04166-1). This work also studied cellulose hydro- and organogels produced by forming a 3D network from cellulose particles. Besides, potentially any cellulose solution can be used for its regeneration, including ones in ZnCl2 aqueous solutions (10.1016/j.carbpol.2021.118946), ZnCl2 molten salt hydrates (10.1007/s10570-022-04712-5), molten hydrates of LiClO4, NaSCN/KSCN/LiSCN and LiCl/ZnCl2 (10.1023/A:1025128028462), and ethylenediamine solutions of NaSCN, KSCN, or NaI (10.1021/acssuschemeng.5b01247).

Line 165: “Ionic lniquid” -> “Ionic liquid”.

Line 186: “and viscosity of cellulose mixture in IL solutions are reduced by inclusion of a cosolvent”. This statement is made without any reference, whereas a detailed study of this issue has been presented recently (see 10.3390/ijms24098057). The same work shows that the DMSO content in IL can be as high as 75%, while Table 1 does not provide information on the possibility of DMSO content above 60%. In addition, the cellulose nanofiltration membranes discussed in section "4.2. Membranes for water treatments" were studied in this work.

Line 469: “to prepare cellulose hydrogels”. Besides cellulose hydrogels, the review examines connected products such as electrochemical materials and membranes. Therefore, the following phrase is more appropriate here ~ “to prepare cellulose hydrogels and related products”.

Round 2

Reviewer 2 Report

None.

Reviewer 3 Report

The author has greatly expanded and improved the review article. It can now be published.